# SciAgent OLab: Open Multi-Agent LLMs for Accelerating Biomedical Discovery

Suk Yee Yong[1] , Rob Dunne[2] , Carol Lee[3] , and Alex McAuley[4]

[1] Information Management & Technology Scientific Computing, Commonwealth Scientific and Industrial Research Organisation, Eveleigh, NSW 2015, Australia
sukyee.yong@csiro.au
[2] Data61, Commonwealth Scientific and Industrial Research Organisation, Sydney, NSW 2015, Australia
[3] Australian e-Health Research Centre, Commonwealth Scientific and Industrial Research Organisation, Sydney, NSW 2145, Australia
[4] Australian Centre for Disease Preparedness (ACDP), Commonwealth Scientific and Industrial Research Organisation (CSIRO), VIC 3219, Australia

**Abstract.** Large language models (LLMs) combined with agentic AI frameworks are increasingly applied to accelerate biomedical research, offering new opportunities for generating insights that can guide real-world experimental studies. Existing tools such as Virtual Lab [4] exemplify this approach but rely on proprietary, cloud-hosted LLMs, creating barriers related to privacy and cost. We present Science Agent Open Lab (SciAgent OLab), a framework that delivers Virtual Lab functionality using locally deployed, open-weight LLMs while maintaining multi-agent collaboration. SciAgent OLab coordinates role-specialized agents to automate literature-driven workflows, including query generation, document retrieval, summarization, and structured knowledge extraction, with optional human oversight for high-level guidance. To demonstrate its utility, we conducted a pilot case study on influenza and SARS-CoV-2 co-infection, where agents curated relevant publications and synthesized key insights to support research exploration. By enabling secure and cost-effective AI-driven workflows without reliance on proprietary services, SciAgent OLab provides a practical framework for accessible and scalable applications in biomedical research.

**Keywords:** Agentic AI · Large Language Model· Multi-Agent · Open-weight Models · Influenza · SARS-CoV-2 Co-infection

## 1 Introduction

Agentic AI frameworks have recently shown promise in biological research workflows, as shown by Swanson et al. [4]. Their work introduced Virtual Lab, where multiple large language model (LLM) agents collaborate to design and validate scientific hypotheses and experiments. Notably, Virtual Lab demonstrated AI-driven nanobody design for SARS-CoV-2 with experimentally confirmed candidates, underscoring its potential for real-world biomedical discovery. However,

its reliance on proprietary, cloud-based LLMs raises concerns around privacy and cost that limit accessibility for many research groups.

SciAgent OLab addresses these limitations by providing a local alternative using open-weight LLMs with multi-agent coordination. This removes proprietary dependencies and improves autonomy and accessibility. Our initial case study applies SciAgent OLab to investigate influenza and SARS-CoV-2 co-infection, aiming to generate hypotheses and identify disrupted pathways for further experimental validation.

## 2 Methodology

SciAgent OLab adopts a team-based structure from Virtual Lab [4], where collaboration occurs through team meetings to share progress and refine strategies and individual meetings for specialized tasks. The virtual team is coordinated by a principal investigator agent for decision-making, specialist agents for domain-specific tasks, and a critic agent for quality assurance, with a human researcher providing high-level guidance when needed.

Agents coordinate to automate a literature-driven pipeline:

Step 1: **Query generation**: LLM agent formulates PubMed queries for the research agenda.

Step 2: **Literature retrieval and triage**: For each query, the agent uses PubMed search [2] to retrieve a set of articles and selects those most relevant to the topic.

Step 3: **Metadata extraction**: Extract PubMed Central Identifier (PMCID) and titles of selected papers.

Step 4: **Full-text download**: Retrieve papers from PubMed Central as JSON files [2] using PMCID.

Step 5: **Summarisation**: LLM agent summarizes each paper, focusing on objectives, methods, and key findings.

Step 6: **Dataset preparation**: Summaries are structured for fine-tuning.

Step 7: **(Planned) Fine-tuning**: Fine-tune open LLM and evaluate performance.

## 3 Experiment on Co-Infection Analysis

### 3.1 Setup

We conducted a pilot study using SciAgent OLab to explore synergistic and antagonistic interactions during influenza and SARS-CoV-2 co-infection. Agents and roles are depicted in Fig. 1.

- **Goal**: Identify disrupted pathways and potential antiviral therapies strategies to reduce severity and impact of co-infection using multi-omics data.
- **Agenda**: Understanding antiviral immune response during SARS-CoV-2 and influenza co-infection.

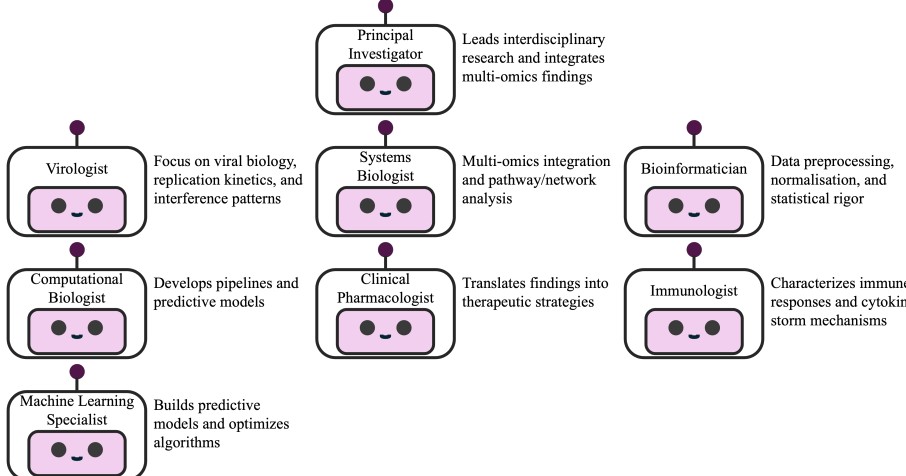

**Fig. 1.** Design of agents and roles in SciAgent OLab for co-infection analysis.

Large model parameter LLMs generally provide better reasoning; however, they require significant memory and compute resources. To balance accuracy and resource constraints:

- For Steps 1–2 (query generation and retrieval), we use a large open-source model `gpt-oss-120b` [3].
- For Steps 5–6 (summarization and dataset preparation), we use a smaller model `phi-4` [1] with $\sim$14 B parameters that supports concurrent threads and offers a long context window (16 K tokens).

This trade-off was validated against the original Virtual Lab experiment on nanobody design, ensuring acceptable accuracy while reducing resource overhead.

### 3.2 Results

We illustrate results from a virologist agent exploring the topic on influenza pathogenesis. Example queries generated by `gpt-oss-120b` [3]:

- influenza virus pathogenesis innate immune responses
- molecular mechanisms of influenza A virus virulence and tissue tropism
- host interferon signaling modulation by influenza viruses

Table 1 lists selected PubMed articles for the first query.
A summary snippet for PMCID 9185163 using `phi-4` [1]:

> **Overview** This comprehensive review focuses on the role of innate lymphoid cells (ILCs)-particularly during influenza infection-and explores their implications for understanding antiviral immune responses amid concurrent SARS-

**Table 1.** Subset of PubMed articles selected for "influenza virus pathogenesis innate immune responses" by virologist agent using `gpt-oss-120b` [3].

| PMCID | Title |
| --- | --- |
| 9185163 | Human Innate Lymphoid Cells in Influenza Infection and Vaccination |
| 6328934 | Influenza virus N-linked glycosylation and innate immunity |
| 10283447 | Pathogenicity and virulence of influenza |

CoV-2 and influenza infections. It highlights the significance of ILCs in maintaining tissue homeostasis, mounting immune defenses, and potentially influencing outcomes in mixed viral infections.

## 4    Discussion and Future Work

SciAgent OLab enables Virtual Lab functionality locally, addressing privacy and cost concerns while preserving agentic coordination. Current progress focuses on extending the framework with fine-tuning and evaluation workflows. Planned experiments include comparing base and fine-tuned models across different families and sizes, assessing performance and factual accuracy, and validating outputs with domain experts to ensure scientific rigor.

Applications include deploying fine-tuned LLMs in interactive interfaces for domain-specific exploration. Potential use cases range from automated literature reviews that extract key insights from large datasets to frameworks that assist researchers in exploring complex topics and generating new hypotheses. Another key direction involves incorporating experimental data to enable end-to-end analysis workflows, bridging computational and experimental research.

Several limitations remain. Running large models locally is compute-intensive, though it may improve reasoning depth and factual accuracy. Context length constraints restrict the number of papers processed per query, which could be mitigated by batching documents. Summarization quality also depends heavily on model capabilities.

## 5    Summary

SciAgent OLab demonstrates how open-weight, locally deployed LLMs can replicate and extend the agentic coordination of Virtual Lab while addressing privacy and cost concerns. Our initial case study on influenza and SARS-CoV-2 co-infection highlights the feasibility of literature-driven pipelines for hypothesis generation. Future work will focus on fine-tuning, rigorous evaluation, and expanding applications such as automated literature reviews and interactive research assistants. SciAgent OLab provides a practical framework for enabling efficient and accessible biomedical discovery.

**Acknowledgments.** The research effort presented in this work was partially funded by the U.S. Food and Drug Administration, Office of the Chief Scientist, Office of Regulatory and Emerging Science Medical Countermeasures Initiative contract (FDA BAA-21-00123 Research Area #7 MCMWP #110). The article reflects the views of the authors and does not represent the views or policies of the FDA.

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
