# OpenReview forum: "SciAgent OLab: Open Multi-Agent LLMs for Accelerating Biomedical Discovery"
_AJCAI/2025/Workshop/AIML-CEB — AIML-CEB 2025 Poster_

### Official Review · Reviewer_MNkw · 2025-11-06
**A nice implementation of an open-weight multi-agent system for biomedical discovery**

**Rating:** 7
**Confidence:** 5

**Review:**

This paper presents a nice open-weight alternative to proprietary-backbone multi-agent biomedical discovery systems.

The implementation seems to still be in the early stages, and it would be nice to eventually have a thorough evaluation of the system and comparison to other baseline systems (e.g. looking at pass@k metrics if possible etc), as the authors mention.

However this seems like a great candidate poster for the workshop, and having this work there will allow the authors to receive timely feedback from the community.

---

### Official Review · Reviewer_VZr8 · 2025-11-07

**Rating:** 7
**Confidence:** 4

**Review:**

This paper presents SciAgent OLab, an open-source alternative to proprietary multi-agent science assistants such as Virtual Lab, designed to enhance accessibility, privacy, and reproducibility in biomedical discovery.

The proposed workflow is logically structured, with clear stages for query generation, retrieval, summarization, and hypothesis construction.

This paper would be a strong fit for the workshop's poster session, where its open-source framework and biomedical focus could stimulate meaningful discussion and interdisciplinary exchange.

**Suggestions for Improvements**

The case study presented is qualitative and small-scale, which effectively demonstrates the feasibility of the approach but lacks quantitative validation. To strengthen the work, the authors should include performance benchmarks comparing their system with proprietary models on tasks such as summarization or information extraction. Additionally, reporting system-level metrics such as latency and resource usage, would substantiate claims regarding efficiency. A discussion on ethical considerations and data governance issues related to deploying LLMs in biomedical discovery would provide a more balanced and responsible perspective.

---

### Official Review · Reviewer_Kz9Q · 2025-11-11
**Could be very interesting if accessible on consumer GPUs**

**Rating:** 7
**Confidence:** 5

**Review:**

This is an interesting look at how smaller LLMs can perform focused tasks as agents, contributing (presumably) to a larger, more comprehensive synthesised body of work. Choice of models for the agents and the actual biological insights generated would be good to see in more detail. Also interesting would be whether guidance on experimental design, time, cost etc can be generated that reflects reality. I am keen to hear about the limitations as well, particularly around document retrieval and accuracy of data extraction, both of which are key challenges in this increasingly crowded field of AI automated/augmented research. Given the scope of the workshop, I hope the authors focus on the validity and usefulness of the biological insights equally in comparison to the agentic framework itself.

---

### Decision · Program_Chairs · 2025-11-12

Accept (Poster)